# Impact of Environmental Tax on Corporate Sustainable Performance: Insights from High-Tech Firms in China

**DOI:** 10.3390/ijerph20010461

**Published:** 2022-12-27

**Authors:** Xiaomin Zhao, Jiahui Li, Yang Li

**Affiliations:** School of Management, Shanghai University, Shanghai 200444, China

**Keywords:** environmental tax, green innovation, sustainable performance

## Abstract

High-tech enterprises play an important role in leading the future industrial transformation, and their sustainable development deserves attention. Using data of 263 high-tech listed firms in China, we explore the impact of environmental tax on corporate sustainable performance, and the mediating role of green innovation. The results show that environmental tax positively affects corporate green innovation. However, the impact of environmental tax on the sustainable performance of state-owned enterprises and private enterprises is quite different. For private enterprises, environmental tax has an inverted U-shaped impact on both corporate financial performance and environmental-social performance. Furthermore, green innovation mediates the relationship of environmental tax and financial performance. In contrast with private enterprises, for state-owned enterprises, environmental tax has a negative linear impact on corporate financial performance. There is no empirical evidence supporting the effect of environmental tax on the environmental social performance of state-owned enterprises. The results imply that the government should implement different tax policies according to the firms’ characteristics, to promote the corporate sustainable development, especially state-owned enterprises.

## 1. Introduction

Environmental concerns and resource limitations have made sustainable development a vital global issue. Environmental protection is necessary for the long-term development of human society as well as business firms [1]. For firms, it is becoming increasingly important to achieve environmental and social performance while creating economic performance [2]. Green innovation has been deemed essential in dealing with environmental issues, as well as an effective strategy to promote a sustainable competitive advantage [3,4].

The study of Centobelli, Cerchione, and Singh [5] showed a significant positive impact of green innovation on financial and environmental performance. Tariq, Badir, and Chonglerttham [6] also supported that eco-product innovation can enhance firms’ profitability and mitigate the financial risk. However, Duque-Grisales, Aguilera-Caracuel, and Guerrero-Villegas [7] revealed that implementing effective green innovation is not associated with greater financial performance. The findings of Baah et al. [8] confirmed a negative and significant influence of green innovation on firms’ financial performance.

The negative relationship indicates that the costs and investments associated with green practices are significant. Such expenditures significantly drain financial resources, especially in the short-term [8]. Financial burden leads to the lack of incentives for firms to carry out green innovation [2]. Farza, Ftiti, Hlioui, Louhichi, and Omri [9] argued that firms rarely prioritize environmental innovation in the absence of institutional pressure, and, thus, government intervention is quite necessary.

As an important intervention mechanism, environmental tax plays a significant role in guiding firms’ green innovation. Reasonable environmental tax can stimulate firms to improve environmental quality, increase energy efficiency, and achieve a win–win situation of economic and environmental benefits [10,11]. However, strict environmental regulation means additional costs, which will crowd out capital investment that could be used for innovation, hinder technological innovation, and reduce firms’ competitiveness [12,13].

Is environmental tax an “incentive effect” or “crowding out effect” on corporate sustainable performance? Will environmental tax promote green innovation? What is the role of green innovation in the impact of environmental tax on corporate sustainable performance? These issues deserve attention.

With the proposal of carbon peaking and carbon neutrality goals, China has entered an important stage of establishing a green, low-carbon, and circular economy. High-tech firms are key powers in technological innovation and green innovation, which play a vital role in China’s economic transformation and high-quality development. As a core part of the future industry, the sustainable performance of high-tech firms is particularly worth studying.

Our paper aims to gain a more comprehensive understanding of the impact of environmental tax on the sustainable performance of high-tech firms. Based on the classical theoretical logic of “external pressure→green behavior→corporate performance”, we first explore the relationship between environmental tax and corporate sustainable performance. Secondly, we analyze the impact of environmental tax on green innovation. Third, we investigate the mediating effect of green innovation on the relationship between environmental tax and corporate sustainable performance. Finally, considering the differences of ownership structure, we further explore the heterogeneity of the impact of environmental tax on state-owned enterprises and private enterprises.

The remaining parts of this paper are organized as follows: Section 2 provides the theoretical foundation and hypothesis. Section 3 describes the sample data and variables. Section 4 provides the empirical research findings and heterogeneity analysis. Section 5 summarizes the main results and offers conclusions.

## 2. Theoretical Foundation and Hypothesis

### 2.1. Corporate Sustainable Performance

Corporate sustainable performance refers to a firm’s capacity to sustain a competitive advantage over time by reducing hazardous emissions and improving green innovation skills [2]. Baah and Jin [14] pointed out that sustainable performance needs to be considered from a triple bottom line: environmental, social, and economic performances, and managers must promote operational activities that achieve economic objectives and achieve both social and environmental goals simultaneously.

Synergies between financial performance and environmental performance allow for a more comprehensive assessment of whether a firm meets the requirements of sustainable development [15]. Creating a balance between economic development and resource consumption is considered a challenge that obliges firms to implement environmentally friendly business activities that improve their financial, social, and environmental performance [16]. Financial performance explains a firm’s ability to create economic benefits to ensure its long-term survival in the market [15]. Social performance refers to the satisfaction of stakeholders, such as the client, end user, government, and the public [17]. Environmental performance represents the achievement of environmental management activities carried out by firms [18]. Therefore, we adopt financial performance and environmental social performance to measure corporate sustainable performance.

### 2.2. Environmental Tax and Corporate Sustainable Performance

#### 2.2.1. Environmental Tax and Corporate Financial Performance

Pigou [19] advocated that the external problems caused by environmental pollution should be transformed into the internal costs of polluters’ emissions by means of taxation. The polluter should be taxed according to the degree of harm caused by pollution, namely the polluter pays principle (PPP) [10].

Some studies show that environmental taxes have a positive impact on corporate financial performance. Zhao and Sun [20] conducted an empirical study on high-polluting firms in China, and the results showed that environmental regulation can promote firms’ economic growth. Berman and Bui [21] found that environmental taxes can impel firms to increase productivity and, thus, promote financial performance.

However, environmental taxes also have a negative impact on firms. Firms are forced to pay more governance costs to solve environmental pollution problems, which crowds out normal productive investment, increases operating costs, and damages corporate profits [22]. According to the study of Steinbrunner [23] on manufacturing firms, upstream energy taxes decrease firms’ technical efficiency, and downstream pollution taxes decrease firms’ productivity, thus limiting financial performance.

Considering the above views, we believe that appropriate environmental taxes can regulate corporate behavior and stimulate corporate innovation, thereby improving financial performance. However, excessive environmental taxes will cause significant cost pressure on firms and reduce financial performance. Therefore, we propose the following hypothesis:

**Hypothesis** **1a.**
*There is an inverted U-shaped nonlinear relationship between environmental tax and financial performance.*


#### 2.2.2. Environmental Tax and Corporate Environmental Social Performance

Regarding the relationship between environmental taxes and corporate environmental social performance, existing studies have also presented two views. Environmental taxes can promote firms to increase environmental R&D investment, develop new environment-friendly products, and establish a green corporate image, and then bring an innovation compensation effect and first-mover advantage effect [10]. Chien et al. found that environmental taxes can positively impact the environment in Asian countries [24], and they think environmental taxes are helpful to reduce pollutant emissions, optimize energy structure, and improve the quality of the ambient air quality and environmental quality [25,26].

However, the green paradox theory holds that unreasonable environmental regulation will accelerate the exploitation and utilization of natural resources and trigger a large number of greenhouse gas emissions [27,28]. Based on neoclassical economics, environmental taxes are the external costs of the use of various natural resources. The current environmental tax policy will make resource owners realize that the cost of exploiting natural resources will become higher and higher in the future, and accelerate energy exploitation in order to reduce costs and improve benefits [29].

Based on the above views, we propose the following hypothesis:

**Hypothesis** **1b.**
*There is an inverted U-shaped nonlinear relationship between environmental tax and corporate environmental social performance.*


### 2.3. Environmental Tax and Corporate Green Innovation

Environmental regulation can encourage firms to increase innovation input, apply external knowledge, and implement an environmental management system [30]. Zhao et al. [12] found that environmental taxes can promote the digestion, absorption, and application of new technologies and provide conditions for high-quality innovation.

According to Yu, Zhang, and Bi [31], environmental tax can motivate firms to carry out green innovation. On the one hand, environmental tax promotes green product innovation. Under the objective of market competition and value maximization, firms will choose to eliminate non-green products and turn to green product innovation, to reduce environmental tax burden and enhance firms’ competitiveness. On the other hand, environmental tax promotes green process innovation, encouraging firms to use green energy and raw materials, reduce energy consumption, reduce emissions, and realize green process innovation [4,8].

Based on the above studies, we hypothesize:

**Hypothesis** **2.**
*Environmental tax is positively correlated to corporate green innovation.*


### 2.4. The Mediating Role of Corporate Green Innovation

The impact of environmental regulation on corporate performance cannot be separated from the role of firms’ innovation practices [13]. The Porter hypothesis points out that environmental regulation promotes firms to carry out more innovative activities, and these innovations improve firms’ productivity, thereby offsetting the cost brought by environmental protection and enhancing firms’ profitability in the market [32]. The findings of Lei et al. [11] support that environmental taxes can improve corporate performance by bringing innovation compensation, to achieve a strong Porter effect.

Environmental taxes help to drive firms with low productivity out of the market, forcing firms to carry out green innovation and resource reallocation [33]. When firms conduct green innovation, they can accumulate knowledge resources, thereby improving financial performance [4,34], reducing environmental pollution, reducing energy consumption, and achieving better environmental and social performance [15,31,35].

Based on the above views, we believe that the impact of environmental tax on corporate financial performance and environmental social performance has a transmission mechanism, in which green innovation plays a mediating role. Therefore, we propose the following hypotheses:

**Hypothesis** **3a.**
*Green innovation mediates the relationship between environmental tax and corporate financial performance.*


**Hypothesis** **3b.**
*Green innovation mediates the relationship between environmental tax and corporate environmental social performance.*


The conceptual model is shown in Figure 1.

## 3. Data and Variables

### 3.1. Data

Our study took high-tech firms in China as a sample. We initially chose 645 high-tech listed firms from the RESSET Database between 2015 and 2019. Considering the availability and universality of the data, this study was screened as follows. First, we eliminated samples that went public after 2015. Second, we eliminated samples that were delisted or listed as special treatment (ST) and particular transfer (PT) during the sample period. Third, we eliminated samples with missing key variables and control variables. Finally, we obtained a dataset of 263 listed firms.

The data of high-tech listed companies came from the RESSET database, which is a data platform that provides professional services for model checking and investment research. Data sources for this study were as follows: (a) environmental taxes, financial performance, and control variables came from the China Stock Market & Accounting Research (CSMAR) Database. The CSMAR Database is a reliable research database covering China’s securities, futures, foreign exchange, macro, industry, and other major economic and financial fields. (b) Corporate green innovation came from the Chinese Research Data Services (CNRDS) Platform, which is a high-quality, open, and platform-based comprehensive data platform for Chinese economic, financial, and business research. (c) Environmental social performance came from the total score of corporate environmental–social responsibility released by the Hexun Website, which is a professional financial investment platform and financial information website providing multi-level financial information and trading services. At the same time, to control the influence of outliers, all continuous variables were curtailed by 1% above and below.

### 3.2. Variables

#### 3.2.1. Environmental Tax

According to the OECD’s definition, environmental taxes are an important instrument for governments to shape relative prices of goods and services. Following Yu and Cheng [35], we adopted the sum of environmental-protection-related taxes and fees to measure environmental taxes, including pollution discharge fees, taxes for maintaining and building cities, farmland use taxes, water conservancy construction funds, resource taxes, and river management fees. Considering the endogenous effect of the model and the time lag of environmental tax, we used the one-period lagged environmental tax [36].

#### 3.2.2. Corporate Sustainable Performance

According to Alexopoulos, Kounetas, and Tzelepis [37], and Xie and Zhu [15], we measured corporate sustainable performance from two dimensions: financial performance and environmental social performance. We used ROA (Return on assets) to measure financial performance, which was commonly used in the green innovation literature [4]. About the environmental social performance, we chose the total score of corporate environmental–social responsibility published by the Hexun Website [15], and used Z-score normalization of the data.

#### 3.2.3. Corporate Green Innovation

Green innovation can be measured by indicators such as green R&D investment [38], eco-product certification [39], ISO14001 certification [40], and the number of green patents [41]. According to Lindman and Sderholm [42], as patent counts measure the outcome of the technological development process, researchers have increasingly relied on patent application counts as one of the most important indicators of innovations. Therefore, we used the number of green patent applications to measure green innovation, including green invention patents and green utility model patents.

#### 3.2.4. Control Variables

We controlled the following variables that may influence corporate sustainable performance: (a) Firm size. A firm’s size may have a significant impact on its performance [43], and larger firms may receive more resources to improve their sustainable performance. Referring to Lin, Zeng, Ma, and Chen [44], we used the natural logarithm of total assets to measure it. (b) Firm age. Firms may lose their ability to compete and innovate over time [4]. Following Lin et al. [44], we measured firm age by using the number of years that company has been listed in the Chinese Stock Market at the end of the reporting year. (c) TOA. The higher the profit, the more conducive to green innovation of the firms [45], affecting their performance. The ratio of operating income to total assets was used to measure it [46]. (d) Asset–liability ratio. This reflects a firm’s financial constraints, indicating the effect of a firm’s financial structure and resources [4].

The specific measurement methods of the main variables are shown in Table 1.

### 3.3. Model Specification

To test the hypotheses, we constructed the models shown in Equations (1) to (5). The first two models examine the impact of environmental tax on corporate sustainable performance. The third model examines the impact of environmental tax on green innovation. The last two models examine the mediating role of green innovation between environmental tax and corporate sustainable performance.
*FP_it_ = α_10_ + β_11_ET_it−1_ + β_12_ET_it−1_^2^ + β_13_Controls_it_ +* *ε_it_.*(1)
*ESP_it_ = α_20_ + β_21_ET_it−1_ + β_22_ET_it−1_^2^ + β_23_Controls_it_ +* *ε_it_.*(2)
*GI_it_ = α_30_ + β_31_ET_it−1_ + β_32_Controls_it_ +* *ε_it_.*(3)
*FP_it_ = α_40_ + β_41_ET_it−1_ + β_42_GI_it_ + β_43_GI_it_^2^ + β_44_Controls_it_ +* *ε_it_.*(4)
*ESP_it_ = α_50_ + β_51_ET_it−1_ + β_52_GI_it_ + β_53_GI_it_^2^ + β_54_Controls_it_ +* *ε_it_.*(5)

In these equations, *i* represents the firm; *t* is the year; *α* is the constant; *β* represents the coefficient; *FP* represents corporate financial performance; *ESP* represents corporate environmental social performance; *ET* denotes environmental tax; *ET*^2^ denotes the square of environmental tax; *GI* stands for corporate green innovation; *GI*^2^ stands for the square of corporate green innovation. Further, *Controls* stands for the control variables, including firm size, firm age, TOA, and asset–liability ratio, whereas *ε* is a normal error term.

According to the requirements of the panel data regression analysis, the model selection of each model should be carried out first. When the Hausman test was conducted on the model, the results showed that the fixed-effect model is superior to the random effect model. In addition, as the data structure is balanced panel data and the number of cross-sections is greater than the number of time series, the fixed-effect model was used for analysis, and Stata 16.0 software, a statistical analysis software for analyzing and managing data, was used to test the hypotheses.

## 4. Results

### 4.1. Descriptive Statistics and Correlation Analysis

Table 2 provides the descriptive statistics and Pearson correlation coefficients of the variables. The results show that environmental tax is significantly correlated with corporate green innovation (*r* = 0.380, *p* < 0.01), corporate financial performance (*r* = 0.117, *p* < 0.01), and corporate environmental social performance (*r* = 0.175, *p* < 0.01).

### 4.2. Regression Analysis

The direct effect of environmental tax on corporate sustainable performance is shown in Table 3. Models 3–5 were used to test the impact of environmental tax on financial performance, and models 7–9 were used to test the impact of environmental tax on environmental social performance.

Model 5 demonstrates that the primary coefficient of environmental tax does not pass the significance test, but the quadratic coefficient of environmental tax is negative and significant (*β* = −0.003, *p* < 0.01), indicating that there is an inverted U-shaped relationship between environmental tax and corporate financial performance, thus providing support for H1a. Similarly, as shown in Model 9, there is also an inverted U-shaped relationship between environmental tax and corporate environmental social performance (*β* = −0.025, *p* < 0.01). Thus, H1b is supported.

There is an inflection point in the inverted U-shaped relationship, that is, with the increase in environmental tax, the financial performance and environmental social performance first increase and then decrease, reaching the highest level at the inflection point. We calculated the inflection point, as shown in Figure 2. When the environmental tax is 13.53, the corresponding corporate financial performance reaches the highest level. When the environmental tax is 15.77, the corresponding corporate environmental social performance reaches the highest level. Sample data show that the median of environmental tax is 15.7, which indicates that the current environmental tax policy is suitable for environmental social performance, but it is on the high side for improving corporate financial performance.

The relationship between environmental tax and corporate green innovation is shown in Table 3. Model 2 shows that environmental tax has a significant positive effect on green innovation (*β* = 0.170, *p* < 0.01). H2 is supported, that is, environmental tax can promote corporate green innovation.

Finally, we examined the mediating effect of corporate green innovation. On one side, environmental tax significantly affects corporate sustainable performance (see Model 5 and Model 9 in Table 3), and has a significant positive effect on corporate green innovation (see Model 2 in Table 3). On the other side, the square of corporate green innovation *GI*^2^ has a significant negative effect on corporate financial performance (*β* = −0.003, *p* < 0.05, see Model 6 in Table 3). Further, the non-parametric percentile Bootstrap method with bias correction was used for the test, and the sample size was set to 5000. At the 95% confidence level, the test results of the mediating effect are shown in Table 4. The lower limit and upper limit of the confidence interval of the mediating effect are −0.00115 and −0.00001, respectively, excluding the 0 value, indicating that the indirect effect is significant. Thus, it can be judged that there is a mediating effect [47,48], that is, corporate green innovation mediates the relationship between environmental tax and corporate financial performance, and H3a is supported.

However, corporate green innovation does not have a significant effect on environmental social performance (see Model 10 in Table 3). In the Bootstrap test (see Table 4), the results show that the lower limit and upper limit of the confidence interval of the mediating effect are −0.00244 and 0.00745, respectively, including the 0 value, indicating that the indirect effect is not significant. Therefore, the mediating effect of corporate green innovation between environmental tax and corporate environmental social performance does not pass the significance test, that is, H3b is not supported.

### 4.3. Heterogeneity Analysis

Firms can be divided into state-owned enterprises and private enterprises according to the ownership structure. There are differences in resource endowment and goal orientation between state-owned enterprises and private enterprises, which are highlighted in policy implementation and environmental responsibility undertaking [36]. In order to analyze how environmental taxes affect corporate sustainable performance under different ownership structures, we divided the whole sample into state-owned enterprises (50 enterprises) and private enterprises (213 enterprises) and conducted group regression.

The results in Table 5 and Table 6 show that, for state-owned enterprises, there is a negative linear relationship between environmental tax and financial performance (*β* = −0.011, *p* < 0.1), while the nonlinear relationship (*β* = −0.001, *p* = 0.379) fails to pass the significance test (see Model 5 in Table 5). For private enterprises, there is an inverted U-shaped relationship between environmental tax and financial performance (*β* = −0.004, *p* < 0.01, Model 5 in Table 6). The results show that environmental tax negatively affects the financial performance of state-owned enterprises; the higher the environmental tax, the worse the financial performance of state-owned enterprises. However, the environmental tax has a nonlinear effect on the financial performance of private enterprises. When the environmental tax is less than the inflection point, the levy of environmental tax will help improve the financial performance of private enterprises. When the environmental tax is greater than the inflection point, its levy will have an adverse effect on the financial performance of private enterprises.

In addition, for state-owned enterprises, the linear relationship (*β* = −0.133, *p* = 0.156) and nonlinear relationship (*β* = 0.007, *p* = 0.689) between environmental tax and corporate environmental social performance fail the significance test (Model 9 in Table 5). For private enterprises, there is an inverted U-shaped relationship between environmental tax and corporate environmental social performance (*β* = −0.03, *p* < 0.05, Model 9 in Table 6). The results indicate that environmental tax can promote the environmental social performance of private enterprises, but there is an inflection point; too high environmental tax will have a negative impact. However, the impact of environmental tax on the environmental social performance of state-owned enterprises has not been statistically significantly verified.

Finally, the results in Table 5 and Table 6 show that for private enterprises, corporate green innovation has a significant mediating effect between environmental tax and financial performance (*β* = −0.004, *p* < 0.05, Model 6 in Table 6). However, for state-owned enterprises, the mediating effect of corporate green innovation does not pass the significance test. This implies that environmental tax has encouraged green innovation of state-owned enterprises, but green innovation has not played its role in improving the sustainable performance of state-owned enterprises.

## 5. Discussions and Conclusions

### 5.1. Conclusions

In this study, we take 263 high-tech listed firms in China as samples to investigate the relationship of environmental tax, corporate green innovation, and sustainable performance. The empirical results reveal the following findings:

First, there is an inverted U-shaped nonlinear relationship between environmental tax and corporate sustainable performance. Environmental tax is beneficial to improve the sustainable performance, but too high environmental tax will have negative effects. Moreover, according to the analysis of the inflection point, we find that the current environmental tax policy in China is on the high side for improving corporate financial performance, but it is helpful for corporate environmental social performance.

Secondly, environmental tax can promote green innovation, and green innovation mediates the impact of environmental tax on corporate financial performance, which is consistent with the research results of Fan and Sun [34]. However, the mediating effect of corporate green innovation on environmental tax and environmental social performance has not been verified. One possible reason is the “double externality” problem of green innovation, which causes a lot of uncertainties, and it is difficult to evaluate the influence of green innovation on environmental social performance in the short term. Longer period data should be used for verification in the future.

Finally, compared with state-owned enterprises, the impact of environmental tax on private enterprises is more significant. There is an inverted U-shaped relationship between environmental tax and the sustainable performance of private enterprises, that is, environmental tax plays a role in promoting the sustainable performance of private enterprises, but there is an inflection point, and excessive environmental tax will have a negative impact on the sustainable performance of private enterprises.

As for state-owned enterprises, the data only show a negative and significant linear relationship between environmental tax and financial performance. Unexpectedly, there is no empirical evidence supporting the effect of environmental tax on the environmental social performance of state-owned enterprises. Environmental tax has not played a role in guiding the sustainable development of state-owned enterprises, which deserves further research. In our view, it is necessary for the government to implement different tax policies according to the industry characteristics and the corporate ownership structure, to promote the corporate sustainable development, especially state-owned enterprises.

### 5.2. Managerial Implications

Our study shows that the existing environmental tax policy is appropriate for improving the environmental social performance of private enterprises, although the tax policy causes a burden on their financial performance to a certain extent. Therefore, for private enterprises, they should pay more attention to their resource investment and environmental costs, developing complete green innovation programs including cleaner recycling methods and environmental management systems, to reduce costs and improve financial performance.

For state-owned enterprises, environmental tax promotes their green innovation behavior. However, green innovation has not improved their environmental and social performance, and environmental tax only has a negative impact on their financial performance. In our opinion, it is necessary for the government to strengthen the environmental supervision on state-owned enterprises. A feasible measure is to require the state-owned enterprises to disclose their environmental, social, and governance (ESG) information. According to the study of Veltri et al. (2020), publishing nonfinancial information (NFI) regarding society and the environment is helpful, as mandatory NFI disclosure positively affects the market firm value [49]. Therefore, we suggest that the ESG rating should be regarded as an important evaluation index for obtaining government support, to promote the sustainable development of state-owned enterprises.

### 5.3. Limitations and Further Research Directions

This study explores the nonlinear relationship between environmental tax and corporate sustainable performance, and analyzes the mediating role of green innovation. The empirical results provide important insights into guiding sustainable development in transition economies. However, there are limitations that need to be addressed.

First, corporate green innovation covers two dimensions: input and output [36]. However, under the framework of financial report disclosure, corporate green innovation input is difficult to separate from firms’ R&D input, so we only focus on output index. Future research should seek more data and consider input indicators. Second, limited by the data availability and the industry characteristics, the sample of listed state-owned enterprises in the heterogeneity analysis is small, and the conclusion is not universal. Future studies can expand the sample size to improve the reliability of the conclusions. Third, this study only analyzes the heterogeneous impact of environmental taxes based on corporate ownership structure, but there are other boundary conditions in practice. Future research can explore other contingency factors, such as environmental dynamics [50], corporate governance, and technological development [9].

In spite of these limitations, our research is crucial for both firms and government, as the desire to protect the earth is unabated.

## Figures and Tables

**Figure 1 ijerph-20-00461-f001:**
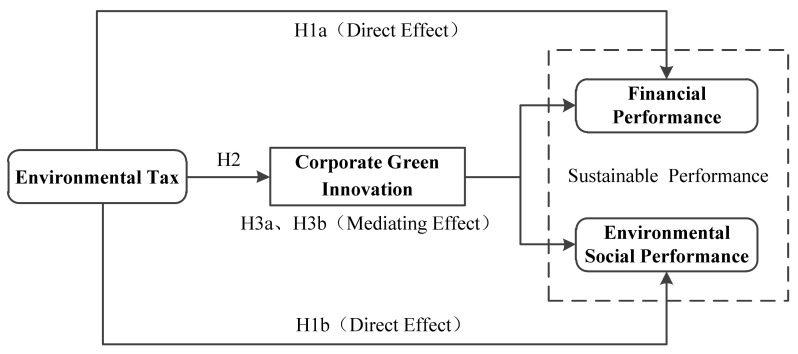
Conceptual model.

**Figure 2 ijerph-20-00461-f002:**
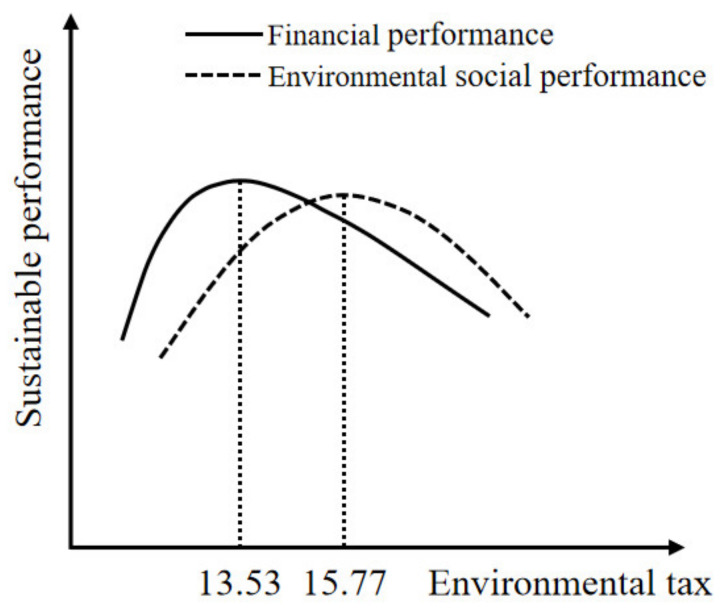
Environmental tax and corporate sustainable performance.

**Table 1 ijerph-20-00461-t001:** Variables and measurements.

Variables	Measures	Literature Sources	Data Sources
Environmental tax (ET)	The natural logarithm of corporate environmental tax plus one	Yu and Cheng [35]	CSMAR Database
Green innovation (GI)	The natural logarithm of the number of applications for green invention patents and green utility model patents plus one	Lindman and Soderholm [42]	CNRDS Platform
Financial performance (FP)	Return on assets (ROA)	Xie et al. [4] Xie and Zhu [15]	CSMAR Database
Environmental Social performance (ESP)	Standardized results of total score of corporate environmental–social responsibility	Xie and Zhu [15]	Hexun Website
Firm size (Size)	The natural logarithm of total assets	Lin et al. [44]	CSMAR Database
Firm age (Age)	The number of years listed in the Chinese Stock Market at the end of the reporting year	Lin et al. [44]	CSMAR Database
TOA	The ratio of operating income to total assets	Lucas and Noordewier [46]	CSMAR Database
Asset-liability ratio (Lev)	The ratio of total liabilities to total assets	Xie et al. [4]	CSMAR Database

**Table 2 ijerph-20-00461-t002:** Descriptive statistics and Pearson correlation coefficients.

Variables	Mean	S.D.	1	2	3	4	5	6	7	8
1. ET	15.84	1.58	1.000							
2. GI	1.34	1.28	0.380 ***	1.000						
3. FP	0.02	0.15	0.117 ***	0.089 ***	1.000					
4. ESP	−0.01	1.00	0.175 ***	0.074 ***	0.492 ***	1.000				
5. Age	12.13	6.32	0.339 ***	0.069 **	−0.064 **	0.036	1.000			
6. Size	22.42	1.15	0.782 ***	0.401 ***	0.061 **	0.202 ***	0.404 ***	1.000		
7. TOA	0.61	0.50	0.301 ***	−0.016	0.071 **	0.085 ***	0.191 ***	0.188 ***	1.000	
8. Lev	0.42	0.21	0.435 ***	0.178 ***	−0.227***	−0.035	0.310 ***	0.593 ***	0.211 ***	1.000

Note: *N* = 263; * *p* < 0.1; ** *p* < 0.05; *** *p* < 0.01; S.D.: Standard deviation.

**Table 3 ijerph-20-00461-t003:** Regression results.

Explained Variables	Green Innovation (GI)		Financial Performance (FP)				Environmental Social Performance (ESP)			
	Model 1	Model 2	Model 3	Model 4	Model 5	Model 6	Model 7	Model 8	Model 9	Model 10
Explanatory variables										
L.ET		0.170 *** (0.047)		−0.007 * (0.004)	−0.005 (0.004)	−0.007 * (0.004)		−0.036 (0.036)	−0.021 (0.035)	−0.038 (0.037)
L.ET^2^					−0.003 *** (0.001)				−0.025 *** (0.009)	
Mediators										
GI						0.006 ** (0.003)				0.002 (0.028)
GI^2^						−0.003 ** (0.001)				0.005 (0.013)
Controls										
Size	0.632 *** (0.038)	0.502 *** (0.060)	0.024 *** (0.003)	0.033 *** (0.005)	0.034 *** (0.005)	0.031 *** (0.006)	0.329 *** (0.033)	0.337 *** (0.047)	0.355 *** (0.047)	0.332 *** (0.049)
Age	0.000 (0.006)	−0.001 (0.007)	−0.001 (0.000)	−0.001 (0.001)	−0.001 (0.001)	−0.001 (0.001)	0.006 (0.005)	0.003 (0.005)	0.004 (0.005)	0.003 (0.005)
TOA	−0.057 (0.085)	−0.167 * (0.096)	0.025 ** (0.010)	0.031 ** (0.013)	0.029 ** (0.013)	0.031 ** (0.013)	0.222 *** (0.068)	0.220 *** (0.078)	0.199 *** (0.076)	0.222 *** (0.079)
Lev	−0.407 ** (0.194)	−0.343 (0.217)	−0.175 *** (0.020)	−0.198 *** (0.023)	−0.200 *** (0.023)	−0.198 *** (0.023)	−1.172 *** (0.154)	−1.257 *** (0.158)	−1.277 *** (0.157)	−1.251 *** (0.159)
Constant	−13.299 *** (0.821)	−12.547 *** (0.958)	−0.438 *** (0.065)	−0.532 *** (0.084)	−0.669 *** (0.124)	−0.604 *** (0.129)	−7.370 *** (0.756)	−6.544 *** (0.757)	−7.429 *** (1.076)	−7.013 *** (1.104)
R^2^	0.397	0.417	0.228	0.247	0.256	0.253	0.273	0.292	0.299	0.292
F-value	18.73	20.84	11.72	10.04	9.42	10.27	10.05	8.4	8.25	8.23
x2	10.30	29.23	109.58	136.41	140.62	136.61	156.32	122.13	125.36	127.25
*p*-value	0.067	0.000	0.000	0.000	0.000	0.000	0.000	0.000	0.000	0.000

Note: Standard errors in parentheses; * *p* < 0.1; ** *p* < 0.05; *** *p* < 0.01.

**Table 4 ijerph-20-00461-t004:** Bootstrap test results of mediating effect.

Explanatory Variable	Mediator	Explained Variable	Bootstrap Test	Effect Size	Boot Standard Error	Boot CI Lower Limit	Boot CI Upper Limit
ET	GI^2^	FP	Mediating effect	−0.00038	0.00028	−0.00115	−0.00001
ET	GI^2^	ESP	Mediating effect	0.00085	0.00232	−0.00244	0.00745

**Table 5 ijerph-20-00461-t005:** Heterogeneity analysis: State-owned enterprises (*N* = 50).

Explained Variables	Green Innovation (GI)		Financial Performance (FP)				Environmental Social Performance (ESP)			
	Model 1	Model 2	Model 3	Model 4	Model 5	Model 6	Model 7	Model 8	Model 9	Model 10
Explanatory variables										
L.ET		0.258 *** (0.079)		−0.011 * (0.006)	−0.011 * (0.006)	−0.009 (0.006)		−0.133 (0.093)	−0.133 (0.093)	−0.147 (0.094)
L.ET^2^					−0.001 (0.001)				0.007 (0.018)	
Mediators										
GI						−0.005 (0.006)				0.033 (0.078)
GI^2^						−0.001 (0.002)				0.006 (0.020)
Controls										
Size	1.073 *** (0.076)	0.786 *** (0.129)	0.031 *** (0.009)	0.039 *** (0.014)	0.039 *** (0.014)	0.043 ** (0.017)	0.268 *** (0.072)	0.390 *** (0.128)	0.385 *** (0.132)	0.355 ** (0.153)
Age	−0.005 (0.017)	0.008 (0.019)	0.003 (0.002)	0.003 (0.002)	0.003 (0.002)	0.003 (0.002)	0.019 (0.020)	0.015 (0.019)	0.015 (0.019)	0.015 (0.020)
TOA	0.158 (0.150)	−0.189 (0.201)	0.004 (0.021)	0.035 (0.024)	0.037 (0.024)	0.034 (0.023)	−0.155 (0.172)	0.039 (0.235)	0.023 (0.242)	0.048 (0.238)
Lev	−0.655 (0.490)	−0.077 (0.545)	−0.179 *** (0.056)	−0.209 *** (0.070)	−0.206 *** (0.069)	−0.212 *** (0.073)	−1.885 *** (0.405)	−2.205 *** (0.502)	−2.227 *** (0.502)	−2.185 *** (0.506)
Constant	−25.088 *** (1.851)	−23.781 *** (2.190)	−0.649 *** (0.195)	−0.599 *** (0.212)	−0.775 *** (0.286)	−0.903 ** (0.374)	−5.049 *** (1.766)	−4.575 ** (2.008)	−6.761 ** (2.746)	−5.808 * (3.449)
R^2^	0.760	0.783	0.375	0.429	0.431	0.432	0.402	0.453	0.453	0.454
F-value	53.95	71.61	12.26	16.86	16.58	15.04	5.46	4.79	4.67	4.64
x2	7.22	4.98	27.60	20.79	24.73	22.94	26.45	24.41	30.07	25.62
*p*-value	0.000	0.000	0.000	0.000	0.000	0.000	0.000	0.000	0.000	0.000

Note: Standard errors in parentheses; * *p* < 0.1; ** *p* < 0.05; *** *p* < 0.01.

**Table 6 ijerph-20-00461-t006:** Heterogeneity analysis: Private enterprises (N = 213).

**Explained Variables**	**Green Innovation (GI)**		**Financial Performance (FP)**				**Environmental Social Performance (ESP)**			
	**Model 1**	**Model 2**	**Model 3**	**Model 4**	**Model 5**	**Model** **6**	**Model** **7**	**Model 8**	**Model 9**	**Model 10**
Explanatory variables										
L.ET		0.212 *** (0.056)		−0.008 (0.005)	−0.007 (0.005)	−0.009 (0.005)		0.009 (0.044)	0.017 (0.042)	0.005 (0.045)
L.ET^2^					−0.004 *** (0.001)				−0.030 ** (0.013)	
Mediators										
GI						0.009 *** (0.003)				0.010 (0.031)
GI^2^						−0.004 ** (0.001)				0.005 (0.017)
Controls										
Size	0.577 *** (0.045)	0.429 *** (0.067)	0.021 *** (0.003)	0.032 *** (0.007)	0.036 *** (0.007)	0.031 *** (0.007)	0.330 *** (0.041)	0.302 *** (0.054)	0.331 *** (0.056)	0.295 *** (0.055)
Age	−0.002 (0.008)	−0.006 (0.008)	−0.001 * (0.001)	−0.001 (0.001)	−0.001 (0.001)	−0.001 * (0.001)	0.005 (0.007)	−0.001 (0.006)	0.000 (0.006)	−0.001 (0.007)
TOA	0.029 (0.108)	−0.093 (0.119)	0.024 * (0.013)	0.026 (0.018)	0.022 (0.017)	0.026 (0.018)	0.178 ** (0.087)	0.164 (0.101)	0.133 (0.097)	0.165 (0.102)
Lev	−0.497 ** (0.231)	−0.451* (0.259)	−0.165 *** (0.023)	−0.194 *** (0.027)	−0.200 *** (0.027)	−0.194 *** (0.027)	−0.939 *** (0.181)	−1.019 *** (0.186)	−1.059 *** (0.185)	−1.009 *** (0.187)
Constant	−12.128 *** (0.955)	−11.528*** (1.075)	−0.388 *** (0.071)	−0.505 *** (0.094)	−0.702 *** (0.152)	−0.590 *** (0.154)	−7.421 *** (0.902)	−6.469 *** (0.904)	−6.873 *** (1.259)	−6.172 *** (1.247)
R^2^	0.340	0.361	0.229	0.242	0.257	0.251	0.278	0.285	0.292	0.285
F-value	15.18	13.96	10.26	8.38	7.94	9.10	9.02	7.25	6.98	7.14
x2	4.96	22.01	86.49	124.45	126.96	121.72	135.58	100.09	106.45	106.67
*p*-value	0.000	0.000	0.000	0.000	0.000	0.000	0.000	0.000	0.000	0.000

Note: Standard errors in parentheses; * *p* < 0.1; ** *p* < 0.05; *** *p* < 0.01.

## Data Availability

All necessary data samples are provided in the paper.

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
