# Peer review of "Impact of Environmental Tax on Corporate Sustainable Performance: Insights from High-Tech Firms in China"

_ijerph, 2022, doi:10.3390/ijerph20010461_

Round 1

Reviewer 1 Report

With the growing concerns on environment, sustainable development has become increasingly important. This paper takes high-tech firms in China as a sample to study the relationship between environmental tax and corporate sustainable performance. The study shows, for private enterprises, environmental tax has an inverted U-shaped impact on both corporate financial performance and corporate environmental social performance. And green innovation mediates the relationship of environmental tax and financial performance. For state-owned enterprises, the study reveals a negative linear impact of environmental tax on financial performance. The results are valuable for policy making. My overall recommendation for this paper is minor revision.

1. Although the paper reads well, syntax, grammar, etc. should be checked to adjust imprecise sentences. For example, in Section 2.1 “The synergy of financial performance and environmental performance can more comprehensively assess …...”, it is difficult to understand and does not explain sustainable performance well.

2. The literature needs to be enhanced. There is a number of research articles published on corporate sustainable performance in the recent years. The authors should include the major articles relevant to the manuscript.

3. The management insights in the conclusion should be further strengthened.

4. In Section 2.2.2, “Environmental taxes can promote firms to increase…, bringing innovation compensation effect and first-mover advantage effect” should be “Environmental taxes can promote firms to increase…, and then bring innovation compensation effect and first-mover advantage effect”. The authors should check such errors throughout the manuscript.

Reviewer 2 Report

I thank the authors for the analysis; below are some considerations.  

Originality: 

The topic covered by the study is certainly interesting and relevant. It is suggested to the authors in the introduction section to emphasise more why the topic is important for publication. In addition, it is also interesting to emphasise more on the gap in the literature. Also, in this section and the literature review, it is suggested to give more reasons why the Chinese context is more interesting than others. In this sense, it will be possible for authors to increase their study's originality and importance further.

Relationship to Literature: 

This section of the study suggests that the authors also include why companies are increasingly interested in sustainable development. In this sense, it will be possible to justify an increased interest on the part of companies in sustainability. It is also interesting to link some aspects of increased corporate interest in financial and non-financial reporting.

In addition, it is also interesting to look into possible regulations in China that have emphasised the interest of companies in sustainable development.

I encourage you to add more references for an extensive Literature review. For example:

-        10.1002/bse.2345

-        10.1007/s10490-012-9291-y

-        10.1016/j.jenvman.2021.113420

-        10.1016/j.energy.2017.11.025

Methodology: 

The methodology part is well-written and organised. However, it is recommended that the authors do more to justify the choice of the analysis database and the software used for analysis.

Results: 

Thanks to the numerous tabs, the results are well written and allow the letter writer a good understanding.

Implications for research, practice and/or society: 

The conclusions are well-written and summarise what was seen earlier in the results section. However, I suggest that the authors emphasise their study's practical and theoretical implications. Also, the discussion section should be expanded to include the conclusion section.

Quality of Communication: 

The paper is almost well-written. However, I would recommend proofreading to the authors.

Overall

The paper is interesting. However, I would suggest implementing the significant elements highlighted before.

Round 2

Reviewer 2 Report

I thank the authors for following some suggestions to improve the study.

Therefore, the paper has improved its originality through an in-depth study of the Chinese context due to the changes made. Furthermore, thanks to the changes in the methodology section, it is now clearer why that particular database was chosen for data collection.

In conclusion, the managerial implications added at a later stage were also greatly appreciated.  

Overall

Following the changes made, the paper is ready for publication.